# Unsaponifiable Matter from Wheat Bran Cultivated in Korea Inhibits Hepatic Lipogenesis by Activating AMPK Pathway

**DOI:** 10.3390/foods12214016

**Published:** 2023-11-03

**Authors:** Minju An, Huijin Heo, Jinhee Park, Heon-Sang Jeong, Younghwa Kim, Junsoo Lee

**Affiliations:** 1Department of Food Science and Biotechnology, Chungbuk National University, Cheongju 28644, Republic of Korea; juju7890@naver.com (M.A.); pltreasure11@gmail.com (H.H.); hsjeong@chungbuk.ac.kr (H.-S.J.); 2Wheat Research Team, National Institute of Crop Science, Rural Development Administration, Wanju 55365, Republic of Korea; pjh237@korea.kr; 3Department of Food Science and Biotechnology, Kyungsung University, Busan 48434, Republic of Korea

**Keywords:** *Triticum aestivum* L., wheat bran, unsaponifiable matter, lipid accumulation, hepatocytes, AMPK pathway

## Abstract

Unsaponifiable matter (USM) from wheat bran, a by-product obtained from wheat milling, is abundant in health-promoting compounds such as phytosterols, tocopherols, policosanols, and alkylresorcinols. This study aimed to examine the effects of USM from the wheat bran of normal and waxy type wheat, Saekeumkang (SKK) and Shinmichal (SMC), on hepatic lipid accumulation in free fatty acid (FFA)-induced hepatocytes and to investigate the cellular mechanism. The total phytochemical contents were 46.562 g/100 g USM and 38.130 g/100 g USM from SKK and SMC, respectively. FFA treatment increased intracellular lipid accumulation by approximately 260% compared to the control group; however, treatment with USM from SKK and SMC significantly attenuated lipid accumulation in the hepatocytes in a dose-dependent manner. Moreover, USM downregulated the expression of lipogenic factors such as fatty acid synthase and sterol regulatory-element-binding protein 1c by approximately 40% compared to the FFA treatment group. Treatment with USM promoted lipolysis and positively regulated the expression of the proteins involved in β-oxidation, including peroxisome proliferator-activated receptor α and its downstream protein, carnitine palmitoyltransferase 1A. Moreover, the blockade of AMPK activation significantly abolished the inhibitory effects of USM on hepatic lipid accumulation. These results indicated that the USM from both SKK and SMC can alleviate lipid accumulation in hepatocytes in an AMPK-dependent manner. Therefore, USM from wheat bran may be useful as a therapeutic intervention for treating metabolic-dysfunction-associated fatty liver disease.

## 1. Introduction

Fatty liver disease is related to metabolic dysfunction. Type 2 diabetes mellitus and obesity often accompany metabolic-dysfunction-associated fatty liver disease (MAFLD) [1]. In a recent study, MAFLD was considered a replacement for the term non-alcoholic fatty liver disease (NAFLD), which emphasizes the contributory role of metabolic dysfunction in the onset of this prevalent liver ailment [2]. Lifestyle changes such as regulation of dietary components and increased physical activity may be beneficial in the initial phase of MAFLD [3]. Therefore, dietary modifications for MAFLD can aid in preventing and controlling this condition.

In MAFLD, excessive lipid accumulation in the liver generates lipotoxins, triggering mitochondrial dysfunction, reactive oxygen species (ROS) overproduction, and endoplasmic reticulum stress [4]. AMP-activated protein kinase (AMPK) plays a significant role in hepatocyte lipid metabolism, glucose homeostasis, and insulin sensitivity [5]. AMPK promotes fatty acid oxidation by suppressing acetyl-CoA carboxylase (ACC) and decreasing malonyl-CoA production. Sterol regulatory element-binding proteins (SREBPs) are pivotal transcription factors in lipid synthesis and regulate the genes associated with fatty acid and triglyceride biosynthesis, such as ACC, glycerol-3-phosphate acyltransferase, fatty acid synthase (FAS), and 3-hydroxy-3-methylglutaryl-CoA reductase [6,7,8]. Previously, the activation of AMPK by polyphenol compounds was shown to protect against hepatic steatosis in mice by suppressing SREBP-1c [9]. Therefore, targeting AMPK signaling pathway activation may be a promising therapeutic approach for the treatment of fatty liver disease.

Wheat (*Triticum aestivum* L.) is an important crop and is the main source of bread, cookies, and noodles [10]. Wheat bran comprises the outermost layer of the wheat kernel and is separated from the endosperm and germ [11]. Wheat bran contains various bioactive compounds, such as vitamin B_6_, vitamin B_9_, vitamin E, alkylresorcinols, and phytosterols, and it plays a key role in contributing to the health benefits of whole grains, such as the reduced risk of colon cancer, diabetes, obesity, and cardiovascular disease [12,13]. Despite the high levels of phytochemicals, wheat bran is typically utilized as animal feed or discarded as waste [14]. Thus, bran can serve as a natural antioxidant and a value-added ingredient in functional food products [15]. Wheat can be categorized into normal and waxy types based on the proportions of amylose and amylopectin in the endosperm starch. Normal-type wheat starch contains approximately 30% amylose and 70% amylopectin, whereas waxy-type wheat starch is composed mostly of amylopectin [16]. The proportions of amylose and amylopectin affect the physicochemical properties of starch and the quality of wheat flour and its final products [17]. Among the Korean wheat cultivars, Saekeumkang (SKK), a normal type, is a new high-yield, high-quality variety used for bread and noodle production in Korea [18]. In addition, the waxy-type wheat cultivar “Shinmichal” (SMC) is cultivated to improve the performance of white wheat flour for bread-making applications [19]. After saponification, unsaponifiable matter (USM) remains insoluble in aqueous solutions and becomes soluble in organic solvents. This substance is rich in policosanols, phytosterols, squalenes, and fat-soluble vitamins and has shown anti-hyperlipidemia and anti-photoaging effects [20,21,22]. Hence, this study aimed to assess the phytochemical composition of the USM derived from normal- and waxy-type wheat bran, namely Saekeumkang and Shinmichal. The hepatoprotective molecular mechanisms of USM derived from wheat bran via the AMPK pathway were also investigated.

## 2. Materials and Methods

### 2.1. Chemicals

Vitamin E standards were purchased from Merck (Darmstadt, Germany). Lutein, zeaxanthin, α-carotene, β-carotene, 5α-cholestane, methyl behenate, campesterol, stigmasterol, β-sitosterol, polycosanols, alkylresorcinol homologs, 3-(4,5-dimethylthiazol-2-yl)-2,5-diphenyltetrazolium bromide (MTT), quercetin, fenofibrate, compound C, Oil red O, dimethyl sulfoxide (DMSO), bovine serum albumin (BSA), sodium oleate, and sodium palmitate were obtained from Sigma Chemical Co. (St. Louis, MO, USA).

### 2.2. Preparation of Unsaponifiable Matter from Wheat Bran

USM was prepared via saponification of the wheat bran according to the method described by Ham et al. [23]. Normal- (SKK) and waxy-type cultivar (SMC) whole wheat were provided by the National Institute of Crop Science (NICS) of the Republic of Korea in 2021. Two voucher specimens, SKK (IT332202) and SMC (IT215851), were deposited in the NICS. The wheat bran was obtained using a Buhler mill (Buhler MLU 202; Buhler Corp., Switzerland). Approximately 3 g of the wheat bran powder was used to prepare USM via saponification. Wheat bran was heated at 100 °C for 30 min to inactivate the lipase enzyme. Wheat bran powder (about 3.0 g) was weighed, and 20 mL of ethanol with 6% pyrogallol was added. After sonication for 5 min, 8 mL of 60% aqueous potassium hydroxide was added. The mixture was flushed with nitrogen gas for 15 s. Then, the wheat bran was saponified in a water bath for 50 min at 70 °C. The solution was shaken every 10 min to ensure a well-mixed extraction. After cooling, 20 mL of 2% NaCl was added. The resulting solution was extracted with 20 mL of ethyl acetate/hexane (15:85, *v*/*v*) three times. The supernatant was then collected, filtered, and evaporated under a vacuum at 40 °C. The resulting residue was dissolved in DMSO to achieve a concentration of 40 mg/mL.

### 2.3. Determination of Vitamin E Content

An HPLC system equipped with a PU-2089 pump, an AS-2055 auto injector, and an FP-2020 fluorescence detector (JASCO Co., Tokyo, Japan) was used for vitamin E analysis. Vitamin E (α-, β-, γ-, δ- tocopherols, and α-, β-, γ-, δ- tocotrienols) content was analyzed on a LiChrosphere^®^ 100 Diol column (250 × 4.6 mm, 5 μm i.d.; Merck, Berlin, Germany) according to a previous study [22]. Briefly, USM was dissolved in hexane and filtered through a 0.45 μm PTFE filter before injection into HPLC. The isocratic mobile phase contained 1.3% isopropanol in *n*-hexane, and the flow rate was 1.0 mL/min. The wavelengths were set at 290 nm for excitation and 330 nm for emission for the identification and quantification of tocopherols and tocotrienols. Tocopherol and tocotrienol peaks were identified by comparing their retention times to those of standards.

### 2.4. Determination of the Carotenoid Content

For carotenoid analysis, a reversed-phase HPLC system equipped with a PU-2089 pump, an AS-2055 auto injector, and a UV-2075 UV/VIS detector (JASCO Co., Tokyo, Japan) was used. The concentrations of carotenoids (lutein, zeaxanthin, α-carotene, and β-carotene) were evaluated using a Vydac 201TP54 column (250 × 4.6 mm, 5 μm, Grace, ND, USA). USM diluted with hexane was filtered through a 0.45 μm PTFE filter before injection into HPLC. Acetonitrile/methanol (70:30, *v*/*v*) was used as a mobile phase solution. The flow rate was 1.0 mL/min. The peak was detected at 450 nm [24].

### 2.5. Determination of the Phytosterol and Policosanol Contents

Phytosterols (campesterol, stigmasterol, and β-sitosterol) and policosanols (docosanol (C22), tetracosanol (C24), hexacosanol (C26), octacosanol (C28), and triacontanol (C30)) were measured using an Agilent 7890A GC (Agilent Technology Inc., Wilmington, NC, USA) with a SAC-5 fused-silica capillary column (30 m × 0.32 mm i.d.; Supelco, Bellefonte, PA, USA) and a flame ionization detector (Agilent Technologies, Palo Alto, CA, USA) [25]. Injector and detector temperatures were 300 °C, and oven temperature was 285 °C. The carrier gas was helium, and the total gas flow rate was 1.0 mL/min. For the analysis of phytosterols, to the USM dissolved in chloroform, we added 50 μL of *N*-methyl-*N*-(trimethylsilyl)trifluoroacetamide (MSTFA) and 50 μL of pyridine, which were incubated at 60 °C for 20 min with shaking for derivatization. Then, the mixture was filtered through a 0.45 μm PTFE filter before injection into GC. 5α-Cholestane was used as the internal standard. For the policosanol analysis, 200 μL of MSTFA was added without pyridine for derivatization.

### 2.6. Determination of the Alkylresorcinol Content

The alkylresorcinol (5-*n*-heptadecylresorcinol (C17:0), 5-*n*-nonadecylresorcinol (C19:0), 5-*n*-heneicosylresorcinol (C21:0), and 5-*n*-tricosylresorcinol (C23:0)) contents were determined using an Agilent 7890A GC (Agilent Technologies) with methyl benzoate as the internal standard [26]. Detection was carried out using a flame ionization detector (Agilent Technologies). The carrier gas was helium, and the total gas flow rate was 1.0 mL/min. The injector and detector temperatures were 250 °C and 350 °C, respectively. Oven temperature was held at 150 °C for 2 min, programmed to rise to 320 °C at a rate of 10 °C/min, and held for 7 min. The trimethylsilyl ether derivates of alkylresorcinols were prepared by adding 100 μL of silylating mixture II, according to Horning (Sigma Chemical Co.), and incubating at 60 °C for 30 min.

### 2.7. Preparation of the Free Fatty Acid-Bovine Serum Albumin Conjugation Mixture

First, a free fatty acid (FFA) solution (4 mM sodium palmitate + 8 mM sodium oleate) was prepared in a 50 mM sodium hydroxide solution at 70 °C [27]. A fatty-acid-free BSA solution (10%) was prepared by dissolving BSA in distilled water at 55 °C. Thereafter, an FFA-BSA conjugation solution (10 mM FFA/1% BSA) was obtained by complexing the appropriate amount of FFA solution with the fatty-acid-free BSA solution for 30 min at 55 °C. After cooling at 25 °C, the solution was filtered.

### 2.8. Cell Culture, Sample Treatment, and Cell Viability Assay

The HepG2 cells were grown in Dulbecco’s modified Eagle’s medium containing 10% FBS and 1% penicillin–streptomycin. The cultures were maintained in a humidified incubator with 5% CO_2_. To evaluate the cytotoxicity, HepG2 cells were seeded in 96-well plates at a density of 1 × 10^4^ cells/mL. After 24 h, the HepG2 cells were treated with the FFA mixture and/or USM (10, 20, and 40 μg/mL) and fenofibrate (10 μM) with FFA (500 μM). In the experiment, to investigate the involvement of the AMPK pathway, compound C (10 μM) was pretreated with cells for 1 h, before FFA, FFA, and USM co-treatment. For the MTT assay, MTT reagent (1 mg/mL) was added to each wells. After a 2 h incubation at 36 °C, the medium was removed, and blue crystal formazan crystals were dissolved in dimethyl sulfoxide and added. Then, the absorbance was measured at 550 nm using an Epoch Microplate Spectrophotometer (BioTek, Inc., Winooski, VT, USA).

### 2.9. Oil Red O Staining

The sample-treated HepG2 cells were rinsed twice with phosphate-buffered saline and stained with an Oil red O working solution as previously described [28]. Stained cells were observed under an Olympus CKX41 microscope (Tokyo, Japan).

### 2.10. Glycerol Release Assay

Free glycerol released into the medium was estimated using a Glycerol Assay Kit (Abcam, Milan, Italy). Briefly, the cell culture supernatants were placed in a new 96-well plate. The free glycerol assay reagent was added to each well. After incubation for 15 min, the released free glycerol was measured at 540 nm using an Epoch Microplate Spectrophotometer (BioTek Inc., Kaysville, UT, USA).

### 2.11. Western Blot Analysis

The HepG2 cells were collected and lysed to conduct Western blot analysis [28]. The blots were developed using chemiluminescence and detected using an image analysis system (WSE 6200 Lumino Graph Chemidoc, Atto, Tokyo, Japan). The intensity of the protein bands was measured using the ATTO CS analyzer 4 (Atto).

### 2.12. Statistical Analysis

Values are expressed as the mean ± standard error (*n* = 2 or more). All data for figures were analyzed using a one-way analysis of variance (ANOVA) followed by Tukey’s post hoc test (GraphPad Prism software version 5, GraphPad Software Inc., La Jolla, CA, USA). The contents of phytochemicals in Table 1 were analyzed using one-way ANOVA, followed by Duncan’s multiple comparison test (SAS version 9.4, SAS Institute Inc., Cary, NC, USA). Significance was set at *p* < 0.05.

## 3. Results and Discussions

### 3.1. Phytochemical Contents of USM

The phytochemical contents of the USM from SKK and SMC were analyzed and are listed in Table 1. Wheat bran USM contained high levels of phytosterols, tocopherols, policosanols, and alkylresorcinols. Among the phytochemical contents examined, β-sitosterol was the most abundant component in the USM of SKK and SMC. The total phytosterol content of SKK was higher than that of SMC. Regarding the carotenoid contents, lutein and zeaxanthin were predominant in the USM, while α-carotene was undetected. The major form of vitamin E isomers was β-tocotrienol in the USM of SKK and SMC, whereas γ-tocotrienol was not detected. The main policosanol in the USM of SKK and SMC was tetracosanol, followed by octacosanol. Alkylresorcinol homologs (C17:0, C19:0, C21:0, and C23:0) were observed in the USM; the C21:0 derivative was the predominant component in the USM of SKK and SMC, which is consistent with previous findings [29]. The total alkylresorcinol content of SKK was 1.7 times higher than that of SMC. Wheat bran provides dietary fiber and a diverse range of biologically active compounds such as alkylresorcinols, tocopherols, and sterols [15,30]. Phytosterol content varies depending on the genetic variation of the wheat, ranging from 0.067 to 0.096 g/100 g dry mass [31]. In addition, β-sitosterol and stigmasterol were shown to decrease the absorption of dietary lipids and led to a decrease in the lipid accumulation in the liver [32]. Detecting the total concentrations of lutein and zeaxanthin is the preferred method for assessing the carotenoid content in wheat and wheat bran [33]. The major policosanol in the bran fraction consisted of wheat varieties grown in Kansas and Oklahoma, constituting C24 [34]. A previous study showed that α-tocopherol protects hepatocytes against liver injury in mouse models [35]. Furthermore, policosanol decreased hepatic oxidative stress by increasing glutathione (GSH) and reducing malondialdehyde (MDA) levels [36]. Alkylresorcinols isolated from wheat bran inhibited hepatic triglyceride accumulation and intestinal cholesterol absorption in obese mice [37]. Therefore, these data imply that wheat bran USM containing these active compounds may attenuate MAFLD.

### 3.2. Effects of USM on Lipid Accumulation and Lipogenesis

In this study, an MTT assay was used to determine the cytotoxicity of the FFA and USM in HepG2 cells. Cell viability was not affected following independent treatment with 500 μM of FFA and various concentrations of USM for 24 h in HepG2 cells (Figure 1A,B). As shown in Figure 1C, co-treatment with 500 μM of FFA and USM for 24 h showed no cytotoxicity in the HepG2 cells.

To examine the effects of USM on lipid accumulation, HepG2 cells were treated with FFA and various concentrations of USM for 24 h. However, treatments with USM from SKK or SMC significantly inhibited FFA-induced fat accumulation (Figure 2A,B). In recent studies, the exposure of HepG2 cells to FFA increased lipid accumulation [38,39,40]. A previous study showed that wheat germ extract suppresses lipid accumulation in palmitic-acid-treated HepG2 cells [41].

The protein expressions of SREBP1c and its target lipogenic enzymes, such as FAS, were evaluated to elucidate the inhibition of lipogenesis-related transcription factors. As shown in Figure 3, FFA treatment enhanced the protein levels of SREBP-1c and FAS. In contrast, treatment with the USM from both SKK and SMC decreased the protein expression of SREBP-1c (Figure 3A) and FAS (Figure 3B) in FFA-stimulated HepG2 cells. SREBPs play key roles as transcription factors that regulate lipogenic gene expression through fatty acids [42]. SREBP-1c is a major regulator of fatty acid synthesis, and its expression is enhanced in NAFLD [43]. However, dietary phytosterols inhibit the expression of SREBP-1c and FAS in HFD-induced hyperglyceridemia [44]. Our results indicate that USM from SKK and SMC reduces lipid accumulation by inhibiting the expression of the lipogenic proteins, including FAS and SREBP-1c, in hepatocytes. Moreover, these findings imply that SREBP-1c and FAS could be targets of USM due to their anti-lipogenic effects on hepatocytes.

### 3.3. Effects of USM on Lipolysis and β-Oxidation

The amount of free glycerol released into the culture medium was determined to assess whether the inhibitory effects of the USM from SKK and SMC on lipid accumulation are associated with lipolysis. As shown in Figure 4A, FFA treatment significantly reduced the amount of glycerol released. However, the USM from SKK and SMC led to the increased extracellular secretion of glycerol molecules into the medium compared to that in the FFA-treated cells. Triglycerides stored within lipid droplets can undergo lipolysis, which breaks them down into free fatty acids and glycerol [45]. A previous study showed that lipolysis decreased in cells incubated with oleic acid [46]. Park et al. reported that oligonol enhances lipolysis by increasing the amount of glycerol released in palmitic acid-treated HepG2 cells [47]. Moreover, β-sitosterol induces lipolytic activity in adipocytes [48]. Therefore, the phytochemicals described in USM may enhance lipolysis.

Additionally, we evaluated the fatty acid oxidative genes of PPARα and its target gene, CPT1A, which is required to uptake fatty acyl-CoA into the mitochondria. According to the results, FFA treatment significantly decreased the expression levels of PPARα and CPT1A (Figure 4B and Figure 4C, respectively). In addition, treatment with the USM from both SKK and SMC dramatically increased the expression of PPARα and CPT1A. PPARα regulates the gene expression related to lipid metabolism such as CPT1A [49]. In a previous study, the protein expression level of PPARα was decreased in the NAFLD group compared to those in the control group [50]. In the same study, increased protein levels of PPARα and CPT1A by camel milk exerted protective effects against high fat-induced liver disease [50]. These results demonstrated that USM stimulates lipolysis and enhances fatty acid oxidation, potentially playing a crucial role in mitigating lipid accumulation in hepatocytes.

### 3.4. Effects of USM on the Activation of the AMPK Signaling Pathway

To investigate the effects of the phosphorylation of AMPK and ACC by USM, hepatocytes were cultured in DMEM containing FFA in the presence or absence of USM from SKK and SMC for 24 h. FFA treatment reduced AMPK and ACC phosphorylation, whereas the USM from both SKK and SMC significantly increased p-AMPK and p-ACC expression (Figure 5A,B). Increased hepatic AMPK activity is expected to ameliorate fatty liver disease via multiple mechanisms. AMPK inhibits fatty acid synthesis by inhibiting the phosphorylation of ACC [51]. Lee et al. showed that FFA treatment downregulates AMPK and ACC phosphorylation; however, the AMPK activation inhibits lipid accumulation in FFA-treated HepG2 cells [52]. AMPK regulates lipid-metabolism-related transcription factors, including SREBP1c [9]. Our previous study demonstrated that AMPK activation inhibits the expression of SREBP-1c and FAS, leading to a reduction in hepatocyte lipid accumulation [28]. This study showed that the activation of AMPK by USM from both SKK and SMC may be involved in ameliorating hepatic lipid accumulation.

To investigate whether the USM from SKK and SMC inhibits lipid accumulation by activating the AMPK pathway, compound C (10 μM), an inhibitor of AMPK, was used and incubated with or without samples for 24 h. Lipid accumulation exhibited a significant reduction in the FFA-stimulated HepG2 cells following treatment with the USM from both SKK and SMC; however, treatment with compound C nullified the inhibitory effects of the USM in the HepG2 cells (Figure 5C,D). This suggests that USM regulates intracellular lipid accumulation through AMPK activation. In an earlier study, AMPK activation decreased lipogenesis by inhibiting SREBP-1c activation, consequently leading to the downregulation of ACC and FAS expression [28]. Additionally, AMPK activates SREBP-1c and SREBP-2, decreasing lipogenesis and lipid accumulation [9].

Then, we investigated whether the USM-mediated effects on lipolysis are associated with the AMPK pathway. As shown in Figure 5E, although the USM from both SKK and SMC significantly increased the released glycerol content in the FFA-treated HepG2 cells, treatment with compound C eliminated the effect of the USM on the extracellular glycerol concentration. AMPK promotes lipolysis via phosphorylation of adipose triglycerides and hormone-sensitive lipases [51]. A previous study showed that black wheat extracts (Arriheuk) regulate lipolysis by increasing glycerol release and free fatty acids via the AMPK/sirtuin 1 (SIRT1) signaling pathway [53]. Moreover, several phytochemical-rich extracts, such as the *Poria cocus* wolf extract and *Toona sinensis* extract, showed cytoprotective effects against hepatic steatosis via AMPK activation [38,40]. Our results indicated that the USM from SKK and SMC induces lipolysis by activating the AMPK signaling pathway in HepG2 cells. These results demonstrated that USM reduces the lipid accumulation by activating the AMPK pathway in hepatocytes.

## 4. Conclusions

In conclusion, our study shows that USM from wheat bran improves lipid metabolism in fatty hepatocytes. The USM from two different wheat cultivars, SKK and SMC, contains abundant phytochemicals such as phytosterols, tocopherols, policosanols, and alkylresorcinols, which may protect against FFA-induced hepatosteatosis in HepG2 cells. The USM from both SKK and SMC significantly reduced hepatic lipid accumulation, possibly via inhibiting lipid synthesis, promoting lipolysis, and upregulating fatty acid oxidative protein expression. Furthermore, the USM activated the AMPK signaling pathway by increasing the phosphorylation of AMPK and ACC. AMPK activation by USM appears to be a key factor in inhibiting lipid accumulation in HepG2 cells. Collectively, there was no difference in activity between SKK and SMC, and both USM showed similar anti-lipogenic activity in HepG2 cells. These findings indicate that the USM extracted from wheat bran has the potential for use in functional food for preventing and treating MAFLD. In addition, further research is ongoing to explore the anti-obesity and antioxidant effects of wheat bran USM in vivo and the wheat bran could serve as a promising functional food ingredient.

## Figures and Tables

**Figure 1 foods-12-04016-f001:**
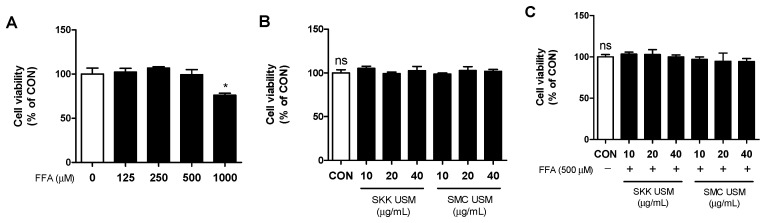
Effects of USM and FFA on cell viability. HepG2 cells were incubated in the indicated concentrations of FFA (**A**) and USM (**B**,**C**) with or without FFA for 24 h. Cell viability was determined using MTT assays. Data are represented as the mean ± standard error (*n* = 3). * *p* < 0.05, significant difference compared with control group. ns, not significant; CON, control; FFA, free fatty acid; USM, unsaponifiable matter; SKK, Saekeumkang; SMC, Shinmichal.

**Figure 2 foods-12-04016-f002:**
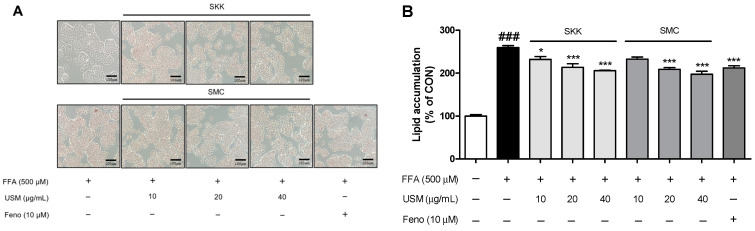
Effects of USM on the lipid accumulation in FFA-induced HepG2 cells. The cells were treated with the indicated concentrations of USM for 24 h, with or without FFA. Fenofibrate (Feno) was used as a positive control. Data are represented as the mean ± standard error (*n* = 3). ^###^ *p* < 0.001 versus the control cells; * *p* < 0.05, and *** *p* < 0.001 versus the FFA-treated group. CON, control; FFA, free fatty acid; USM, unsaponifiable matter; SKK, Saekeumkang; SMC, Shinmichal. (**A**) Microscopic observations of the stained lipid droplets by oil red O staining; (**B**) Quantification of the stained lipid droplets were performed using the eluted Oil red O stain via measuring absorbance at 510 nm.

**Figure 3 foods-12-04016-f003:**
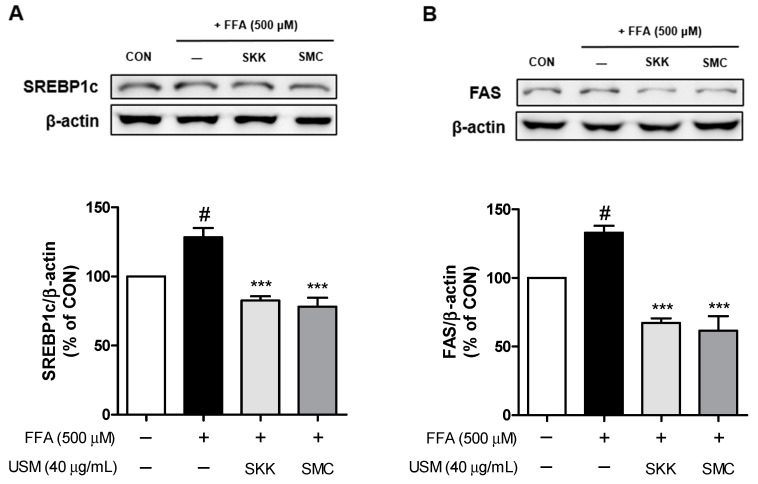
Effects of USM on the expression levels of SREBP1c (**A**) and FAS (**B**) in FFA-induced HepG2 cells. Data are represented as the mean ± standard error (*n* = 3). ^#^ *p* < 0.05 versus the control cells; *** *p* < 0.001 versus the FFA-treated group. CON, control; FFA, free fatty acid; USM, unsaponifiable matter; SREBP1c, sterol regulatory element-binding protein 1c; FAS, fatty acid synthase; SKK, Saekeumkang; SMC, Shinmichal.

**Figure 4 foods-12-04016-f004:**
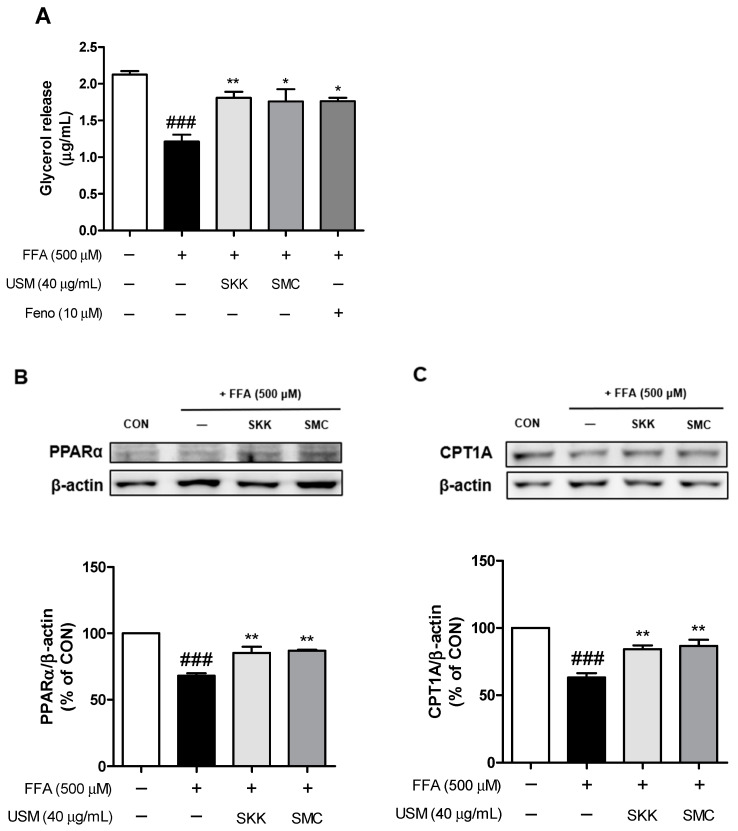
Effects of USM on glycerol release (**A**) and the expression levels of PPARα (**B**) and CPT1A (**C**) in FFA-induced HepG2 cells. Fenofibrate (Feno) was used as a positive control. Data are represented as the mean ± standard error (*n* = 3). ^###^
*p* < 0.001 versus the control cells; * *p* < 0.05, and ** *p* < 0.01 versus the FFA-treated group. CON, control; FFA, free fatty acid; USM, unsaponifiable matter; PPARα, peroxisome proliferator-activated receptor alpha; CPT1A, carnitine palmitoyltransferase 1A; SKK, Saekeumkang; SMC, Shinmichal.

**Figure 5 foods-12-04016-f005:**
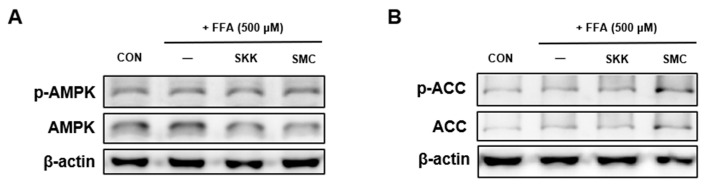
Effects of USM on the expression levels of AMPK (**A**) and ACC (**B**) phosphorylation in FFA-induced HepG2 cells. Involvement of the AMPK pathway in the effects of USM and compound C on the lipid droplets (**C**), lipid accumulation (**D**), and glycerol release (**E**) of FFA-induced HepG2 cells. Data are represented as the mean ± standard error (*n* = 3). ^##^
*p* < 0.01, and ^###^
*p* < 0.001 versus the control cells; ** *p* < 0.01, and *** *p* < 0.001 versus the FFA-treated group; ^$^
*p* < 0.05 ^$$$^
*p* < 0.001 versus the USM-treated group. ns, not significant; CON, control; FFA, free fatty acid; USM, unsaponifiable matter; AMPK, AMP-activated protein kinase; ACC, acetyl-CoA carboxylase; SKK, Saekeumkang; SMC, Shinmichal.

**Table 1 foods-12-04016-t001:** Phytochemical contents of the unsaponifiable matter (USM) from wheat bran.

Phytochemicals	Saekeumkang	Shinmichal
Phytosterol (g/100 g USM)		
Campesterol	5.472 ± 0.429 ^Ab^	4.690 ± 0.135 ^Ab^
Stigmasterol	0.746 ± 0.015 ^Bc^	0.925 ± 0.011 ^Ac^
β-sitosterol	20.447 ± 1.919 ^Aa^	17.752 ± 0.266 ^Aa^
Carotenoid (g/100 g USM)		
Lutein	0.037 ± 0.002 ^Aa^	0.035 ± 0.002 ^Aa^
Zeaxanthin	0.024 ± 0.002 ^Ab^	0.015 ± 0.001 ^Bb^
α-Carotene	ND	ND
β-Carotene	0.003 ± 0.0001 ^Ac^	0.001 ± 0.00004 ^Bc^
Vitamin E (g/100 g USM)		
α-Tocopherol	1.000 ± 0.120 ^Ab^	0.876 ± 0.003 ^Aa^
β-Tocopherol	0.473 ± 0.064 ^Ac^	0.327 ± 0.030 ^Ab^
γ-Tocopherol	0.014 ± 0.001 ^Ad^	0.008 ± 0.0004 ^Bc^
δ-Tocopherol	0.025 ± 0.004 ^Ad^	0.021 ± 0.001 ^Ac^
α-Tocotrienol	0.420 ± 0.063 ^Ac^	0.352 ± 0.013 ^Ab^
β-Tocotrienol	1.406 ± 0.203 ^Aa^	0.890 ± 0.027 ^Aa^
γ-Tocotrienol	ND	ND
δ-Tocotrienol	0.009 ± 0.002 ^Ad^	0.016 ± 0.001 ^Bc^
Policosanol (g/100 g USM)		
Docosanol (C22)	0.095 ± 0.010 ^Bc^	0.150 ± 0.004 ^Ac^
Tetracosanol (C24)	1.681 ± 0.090 ^Aa^	1.523 ± 0.070 ^Aa^
Hexacosanol (C26)	0.099 ± 0.025 ^Bc^	0.213 ± 0.027 ^Ac^
Octacosanol (C28)	1.462 ± 0.144 ^Ab^	1.043 ± 0.019 ^Ab^
Triacontanol (C30)	0.153 ± 0.001 ^Bc^	0.206 ± 0.015 ^Ac^
Alkylresorcinol (g/100 g USM)		
5-Heptadecylresorcinol (C17:0)	0.318 ± 0.01 ^Ac^	0.168 ± 0.012 ^Bc^
5-Nonadecylresorcinol (C19:0)	2.114 ± 0.17 ^Ab^	1.246 ± 0.059 ^Bb^
5-Heneicosylresorcinol (C21:0)	9.464 ± 0.62 ^Aa^	6.668 ± 0.318 ^Ba^
5-Tricosylresorcinol (C23:0)	1.101 ± 0.03 ^Ac^	1.004 ± 0.037 ^Ab^
Total (g/100 g USM)	46.562 ± 3.932	38.130 ± 1.050

Values are represented as the mean ± standard deviation (*n* = 2). Means with different capital letters (A,B) within the same row of each sample are significantly different by Duncan’s multiple range test at *p* < 0.05. Means with different small letters (a–c) within the same column of each sample are significantly different by Duncan’s multiple range test at *p* < 0.05. ND = not detected.

## Data Availability

The data supporting the findings of this study are available from the corresponding author upon reasonable request.

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
