# Peer review of "Unsaponifiable Matter from Wheat Bran Cultivated in Korea Inhibits Hepatic Lipogenesis by Activating AMPK Pathway"

_foods, 2023, doi:10.3390/foods12214016_

Round 1

Reviewer 1 Report

Comments and Suggestions for Authors

In this study, the authors investigate the effects of unsaponifiable matter (USM) from wheat bran on hepatic lipid accumulation in FFA-induced HepG2 cells and the underlying mechanism. The study was well designed and conducted, and the manuscript was well written. The manuscript could after minor revision. My comments are:

1)      Although the authors provided the reference for saponification, they should also briefly describe the saponification process.

2)      The authors should explain how to determine the concentrations (10, 20, and 40 μg/mL) of USM in this study?

3)      In the conclusion, I suggest that the authors might raise questions and identify areas which need further research.

Author Response

1) Although the authors provided the reference for saponification, they should also briefly describe the saponification process.

Answer: We added the brief procedure of saponification in section 2.2 as below.

In section 2.2: Approximately 3 g of the wheat bran powder was used to prepare USM by saponification. Wheat bran was heated at 100 °C for 30 min to inactivate the lipase enzyme. Wheat bran powder (about 3.0 g) was weighed, and 20 mL of ethanol with 6% pyrogallol was added. After sonication for 5 min, 8 mL of 60% aqueous potassium hydroxide was added. The mixture was flushed with nitrogen gas for 15 s. Then, the wheat bran was saponified in a water bath for 50 min at 70°C. The solution was shaken every 10 min to ensure a well-mixed extraction. After cooling, 20 mL of 2% NaCl was added. The resulting solution was extracted with 20 mL of ethyl acetate/hexane (15:85, v/v) three times. The supernatant was then collected, filtered, and evaporated under vacuum at 40°C. The resulting residue was dissolved in DMSO to achieve a concentration of 40 mg/mL.

2) The authors should explain how to determine the concentrations (10, 20, and 40 μg/mL) of USM in this study?

Answer: Our preliminary experiments showed no toxicity at concentrations up to 50 µg/mL, but we observed abnormalities in cell morphology. Therefore, we selected a lower concentration for conducting the experiment safely. Additionally, we conducted experiments at different concentrations, including 10 and 20 µg/mL, to assess concentration-dependent changes in HepG2 cells. We explained the preparation of USM stock solution in DMSO in section 2.2. And, when treating cells with USM, the USM stock solution (40 mg/mL) was diluted with culture medium to achieve concentrations of 40 ug/mL, 20 ug/mL, or 10 ug/mL.

3) In the conclusion, I suggest that the authors might raise questions and identify areas which need further research.

Answer: Thank you for your kind comment. Actually, we are conducting further studies with wheat bran USM to explore its anti-obesity and the antioxidant activities in vivo. Thus, the following sentence has been inserted into the conclusion. “In addition, further research is ongoing to explore the anti-obesity and antioxidant effects of wheat bran USM in vivo and the wheat bran could serve as a promising functional food ingredient.”

Reviewer 2 Report

Comments and Suggestions for Authors

foods-2682519

Include in Keywords tne cientific name  of the plant used   

Kerwords: Triticum aestivum, ......

Line 54 change  folate for vitamin B9

Section 2.2. include  voucher specimen of  plants. Also where were deposited

Section 2.3  was used to determitated vitamin E and carotenoids however must be two different methos in agrement with the text therefore must be 2.3 to vitamin E and 2.4 to carotenoids. Also include if was used any standar or was characterized by library

2.4, and 2.5, section include if was used any standar or library

Table 1  change sitosterol as Sitosterol

Reference

11, 15, 22,25, 27,28,31,38-41,50-55, only use upper case where are neccesary see enclosed yellow color are corrections

Comments on the Quality of English Language

Check bibliography

Author Response

1) Include in Keywords tne cientific name  of the plant used   Kerwords: Triticum aestivum, ......

Answer: We revised the keywords according to your comment.

2) Line 54 change  folate for vitamin B9

Answer: We revised folate to vitamin B9 according to your comment.

3) Section 2.2. include  voucher specimen of  plants. Also where were deposited

Answer: We added following sentence in section 2.2. “Two voucher specimens, SKK (IT332202) and SMC(IT215851) were deposited in NICS.”

4) Section 2.3  was used to determitated vitamin E and carotenoids however must be two different methos in agrement with the text therefore must be 2.3 to vitamin E and 2.4 to carotenoids. Also include if was used any standar or was characterized by library

Answer: We added detailed methods for determination of vitamin E and carotenoid according to your comment as below. Also, standard information was presented in section 2.3 and 2.4.

In section 2.3 and 2.4:

2.3. Determination of vitamin E content

HPLC system equipped with a PU-2089 pump, an AS-2055 auto injector, and an FP-2020 fluorescence detector (JASCO Co., Tokyo, Japan) was used for vitamin E analysis. Vitamin E (α-, β-, γ-, δ- tocopherols, and α-, β-, γ-, δ- tocotrienols) content was analyzed on a LiChrosphere® 100 Diol column (250 × 4.6 mm, 5 μm i.d.; Merck, Berlin, Germany) according to the previous study [22]. Briefly, USM was dissolved in hexane and filtered through a 0.45 μm PTFE filter before injection into HPLC. The isocratic mobile phase contained 1.3% isopropanol in n-hexane and the flow rate was 1.0 mL/min. The wavelengths were set at 290 nm for excitation and 330 nm for emission for the identification and quantification of tocopherols and tocotrienols. Tocopherol and tocotrienol peaks were identified by comparing their retention times to those of standards.

2.4. Determination of the carotenoid content

For carotenoid analysis, a reversed-phase HPLC system equipped with a PU-2089 pump, an AS-2055 auto injector, and a UV-2075 UV/VIS detector (JASCO Co., Tokyo, Japan) was used. The concentrations of carotenoids (lutein, zeaxanthin, α-carotene, and β-carotene) were evaluated using a Vydac 201TP54 column (250 × 4.6 mm, 5 μm, Grace, USA). USM diluted with hexane was filtered through a 0.45 μm PTFE filter before injection into HPLC. The acetonitrile/methanol (70:30, v/v) was used as a mobile phase solution. The flow rate was 1.0 mL/min. The peak was detected at 450 nm [24].

5) 2.4, and 2.5, section include if was used any standar or library

Answer: We used standard chemicals for quantification of policosanols and alkylresorcinol, and all standard information is added in section 2.4 and 2.5.

6) Table 1  change sitosterol as Sitosterol

Answer: We revised it according to your comment.

7) Reference 11, 15, 22,25, 27,28,31,38-41,50-55, only use upper case where are neccesary see enclosed yellow color are corrections

Answer: We revised the it according to your comment. And we found and deleted duplicated references (ref. 54 (=ref. 38) and ref. 55(ref.=40)).

Reviewer 3 Report

Comments and Suggestions for Authors

Review on manuscript: foods-2682519

Unsaponifiable Matters from Wheat Brans Cultivated in Korea Inhibit Hepatic Lipogenesis by Activating AMPK Pathway

 by  Minju An, Huijin Heo, Jinhee Park, Heon Sang Jeong, Younghwa Kim and Junsoo Lee

 submitted to Foods

The manuscript submitted for evaluation the authors studied the the phytochemical composition of the unsaponifiable matters derived from the normal and waxy- type wheat bran and their role in inhibition hepatic lipogenesis by activating AMPK pathway.

In my opinion, the topic taken by the authors is interesting, however, the manuscript should be corrected and supplemented.

 Detailed recommendation:

Abstract – should contain more concrete / numerical results,

line 89 – should be: obtained,

line 93-97 – model, producer and country origin of HPLC system should be given, the detection method should be provided, how were samples prepared for analysis?

lines 100-102 – model of GC system should be given, how were samples prepared for analysis?

line 105 – model, producer and country origin of GC system should be given, the detection method should be provided,

lines 119-125 – if no reference is provided, the methodology used should be described in sufficient detail to enable independent repetition of the experiment,

line 136 – model and country origin should be given,

line 141 – country origin should be added,

Results and discussion – the repetition of numerical data contained in tables should be limited,

Table 1 – the data should be statistically evaluated,

Figure 1 – the data should be statistically evaluated,

Figures 2B, 3B, 4 and 5 – what statistical method was used to evaluate the data? such information should be provided in the methodology,

Conclusion – is it possible to indicate which USM from wheat bran is more effective?

Author Response

1) Abstract – should contain more concrete / numerical results,

Answer: We revised the abstract according to your comment.

2) line 89 – should be: obtained,

Answer: We revised it according to your comment.

3) line 93-97 – model, producer and country origin of HPLC system should be given, the detection method should be provided, how were samples prepared for analysis?

Answer: Following sentences were added in section 2.3 and 2.4.

2.3. Determination of vitamin E content

HPLC system equipped with a PU-2089 pump, an AS-2055 auto injector, and an FP-2020 fluorescence detector (JASCO Co., Tokyo, Japan) was used for vitamin E analysis. Vitamin E (α-, β-, γ-, δ- tocopherols, and α-, β-, γ-, δ- tocotrienols) content was analyzed on a LiChrosphere® 100 Diol column (250 × 4.6 mm, 5 μm i.d.; Merck, Berlin, Germany) according to the previous study [22]. Briefly, USM was dissolved in hexane and filtered through a 0.45 μm PTFE filter before injection into HPLC. The isocratic mobile phase contained 1.3% isopropanol in n-hexane and the flow rate was 1.0 mL/min. The wavelengths were set at 290 nm for excitation and 330 nm for emission for the identification and quantification of tocopherols and tocotrienols. Tocopherol and tocotrienol peaks were identified by comparing their retention times to those of standards.

2.4. Determination of the carotenoid content

For carotenoid analysis, a reversed-phase HPLC system equipped with a PU-2089 pump, an AS-2055 auto injector, and a UV-2075 UV/VIS detector (JASCO Co., Tokyo, Japan) was used. The concentrations of carotenoids (lutein, zeaxanthin, α-carotene, and β-carotene) were evaluated using a Vydac 201TP54 column (250 × 4.6 mm, 5 μm, Grace, USA). USM diluted with hexane was filtered through a 0.45 μm PTFE filter before injection into HPLC. The acetonitrile/methanol (70:30, v/v) was used as a mobile phase solution. The flow rate was 1.0 mL/min. The peak was detected at 450 nm [24].

4) lines 100-102 – model of GC system should be given, how were samples prepared for analysis?

Answer:  Following sentences were added in section 2.5.

Phytosterols (campesterol, stigmasterol, and β-sitosterol) and policosanols (docosanol (C22), tetracosanol (C24), hexacosanol (C26), octacosanol (C28), and triacontanol (C30)) were measured using Agilent 7890A GC (Agilent Technology Inc., Wilmington, NC, USA) with a SAC-5 fused-silica capillary column (30 m × 0.32 mm i.d.; Supelco, Bellefonte, PA, USA) and a flame ionization detector (Agilent Technologies, Palo Alto, CA, USA) [25]. Injector and detector temperatures were 300°C and oven temperature was 285°C. The carrier gas was helium, and the total gas flow rate was 1.0 mL/min. For the analysis of phytosterols, the USM dissolved in chloroform was added with 50 μL of N-methyl-N-(trimethylsilyl)trifluoroacetamide (MSTFA) and 50 μL of pyridine and incubated at 60°C for 20 min with shaking for derivatization. Then, filtered through a 0.45 μm PTFE filter before injection into GC. 5α-Cholestane was used as internal standard. For the policosanol analysis, 200 μL of MSTFA was added without pyridine for derivatization.

5) line 105 – model, producer and country origin of GC system should be given, the detection method should be provided,

Answer:  Following sentences were added in section 2.6.

The alkylresorcinol (5-n-heptadecylresorcinol (C17:0), 5-n-nonadecylresorcinol (C19:0), 5-n-heneicosylresorcinol (C21:0), and 5-n-tricosylresorcinol (C23:0)) content was determined using Agilent 7890A GC (Agilent Technologies) with methyl benzoate as internal standard [26]. Detection was carried out using a flame ionization detector (Agilent Technologies). The carrier gas was helium, and the total gas flow rate was 1.0 mL/min. The injector and detector temperatures were 250°C and 350°C, respectively. Oven temperature was held at 150°C for 2 min, programmed to rise to 320°C at a rate of 10°C /min and held for 7 min. The trimethylsilyl ether derivates of alkylresorcinols were prepared by adding 100 μL of silylating mixture II according to Horning (Sigma Chemical Co.) and incubating at 60°C for 30 min.

6) lines 119-125 – if no reference is provided, the methodology used should be described in sufficient detail to enable independent repetition of the experiment,

Answer: We explained detail methodology for cell culture and MTT assay in section 2.8 as below.

In section 2.8: The HepG2 cells were grown in Dulbecco’s modified Eagle’s medium containing 10% FBS and 1% penicillin-streptomycin. The cultures were maintained in a humidified incubator with 5% CO2. To evaluate the cytotoxicity, HepG2 cells were seeded in 96-well plates at a density of 1´ 104 cells/mL. After 24 hours, the HepG2 cells were treated with the FFA mixture and/or USM (10, 20, and 40 μg/mL) and fenofibrate (10 μM) with FFA (500 μM). In the experiment, to investigate the involvement of AMPK pathway, compound C (10 μM) was pretreated with cells for 1 h, before FFA, FFA, and USM co-treatment. For the MTT assay, MTT reagent (1 mg/mL) was added to each wells. After a 2-hour incubation at 36 °C, the medium was removed, and blue crystal formazan crystals were dissolved in dimethyl sulfoxide and added. Then, the absorbance was measured at 550 nm using a microplate reader (BioTek, Inc., Winooski, VT, USA).

7) line 136 – model and country origin should be given,

Answer: We used an Epoch Microplate Spectrophotometer (Biotek Inc., Winooski, VT, USA). However, the microplate spectrophotometer has already been introduced in section 2.8. Thus, the only model information was shown in section 2.10.

In section 2.8: Then, the absorbance was measured at 550 nm using an Epoch Microplate Spectrophotometer (BioTek, Inc., Winooski, VT, USA).

In section 2.10: After incubation for 15 min, the released free glycerol was measured at 540 nm using an Epoch Microplate Spectrophotometer (BioTek Inc.).

8) line 141 – country origin should be added,

Answer: We used an ATTO image analysis software (CS analyzer 4) from Tokyo, Japan. Thus, we added it In section 2.11.

"The intensity of the protein bands was measured using the ATTO image analysis soft-ware (CS analyzer 4, Tokyo, Japan)."

9) Results and discussion – the repetition of numerical data contained in tables should be limited,

Answer: We revised it according to your comment.

10) Table 1 – the data should be statistically evaluated,

Answer: We added the statistical result in Table 1.

11) Figure 1 – the data should be statistically evaluated,

Answer: Actually, Fig 1 includes the statistical result. Thus, we explained the statistical result in the legend as below.

“* p < 0.05, significant difference compared with control group. ns, not significant”

12) Figures 2B, 3B, 4 and 5 – what statistical method was used to evaluate the data? such information should be provided in the methodology,

Answer: We revised it according to your comment as below.

In section 2.12: “Values are expressed as the mean ± standard error (n = 2 or more). All data for figures were analyzed using a one-way analysis of variance (ANOVA) followed by Tukey’s post-hoc test (GraphPad Prism software version 5, GraphPad Software Inc., La Jolla, CA, USA). Significance was set at p < 0.05.”

13) Conclusion – is it possible to indicate which USM from wheat bran is more effective?

Answer: There was no difference in activity between the two varieties. And, both USM showed similar biological activity in HepG2 cells. We added the following sentence.

In section 4. Conclusion: “Collectively, there was no difference in activity between SKK and SMC, and both USM showed similar anti-lipogenic activity in HepG2 cells.”